# Ab initio nonrigid X-ray nanotomography

Michal Odstrcil [1], Mirko Holler [1], Jörg Raabe [1], Alessandro Sepe[2,3], Xiaoyuan Sheng[3,4], Silvia Vignolini [5], Christian G. Schroer [6,7] & Manuel Guizar-Sicairos[1]

Reaching the full potential of X-ray nanotomography, in particular for biological samples, is limited by many factors, of which one of the most serious is radiation damage. Although sample deformation caused by radiation damage can be partly mitigated by cryogenic protection, it is still present in these conditions and, as we exemplify here using a specimen extracted from scales of the *Cyphochilus* beetle, it will pose a limit to the achievable imaging resolution. We demonstrate a generalized tomographic model, which optimally follows the sample morphological changes and attempts to recover the original sample structure close to the ideal, damage-free reconstruction. Whereas our demonstration was performed using ptychographic X-ray tomography, the method can be adopted for any tomographic imaging modality. Our application demonstrates improved reconstruction quality of radiation-sensitive samples, which will be of increasing relevance with the higher brightness of 4th generation synchrotron sources.

[1] Paul Scherrer Institut, CH-5232, Villigen PSI, Switzerland. [2] Big Data Science Center, Shanghai Synchrotron Radiation Facility, Zhangjiang Laboratory, 239 Zhangheng Road, Pudong, 201204 Shanghai, China. [3] Adolphe Merkle Institute, University of Fribourg, Chemin des Verdiers 4, CH-1700 Fribourg, Switzerland. [4] Department of Physics, University of Cambridge, JJ Thompson Avenue, CB3 0HE, Cambridge, UK. [5] Department of Chemistry, University of Cambridge, Lensfield Road, CB2 1EW, Cambridge, UK. [6] Deutsches Elektronen-Synchrotron DESY, Notkestr. 85, 22607 Hamburg, Germany. [7] Department of Physics, Universität Hamburg, Luruper Chaussee 149, 22761 Hamburg, Germany. Correspondence and requests for materials should be addressed to M.G-S. (email: manuel.guizar-sicairos@psi.ch)

Tomographic reconstruction of time-evolving samples is a challenging but important task that can help to understand dynamic processes inside a sample in a non-destructive way. 4D computed tomography (4D-CT) reconstruction methods[1] provide a significant gain in quality for time-evolving samples compared with simple sequential 3D reconstructions, however, these methods are often based on restrictive prior assumptions, such as periodicity of movement[1–4] or sparsity of the sample and its evolution[4–7]. Using these assumptions, 4D-CT methods can recover the evolution of a dynamic sample, and have been used in studies of cardiac and respiratory motion[1,3,4,8,9].

The generality of the 4D-CT methods, however, comes for the price of increased amount of reconstructed information and therefore needs additional data, such as more tomographic projections or reconstruction constraints. This is a severe limitation in many cases, particularly when the dynamic processes in the sample are not of experimental interest and only lead to deteriorated reconstruction quality without providing any additional information.

We demonstrate here an application of nonrigid geometry computed tomography (NCT) with a self-consistent method for motion estimation directly from the measured data set. Our approach builds on previous work on 4D-CT imaging but reformulates the optimization task in order to avoid the significant increase of the degrees of freedom that would be needed to recover a general 4D-CT time-evolving reconstruction[1,10–12]. In addition, our method accounts for a continuous deformation of the sample structure during data acquisition, i.e., nonrigid sample changes, as well as rigid sample motion that is commonly present due to insufficient stability of nanotomography setups. The goal of our approach is to preserve reconstruction quality comparable with conventional imaging of a static sample without the need of extra information or assumptions, neither about the sample structure nor a particular temporal dependence of the dynamics, such as linear or periodic changes. Our self-consistent approach for estimation of the deformation evolution also removes the requirement of having a high-quality static initial measurement required by some dynamic CT approaches[2,3,10] or need for reference markers. Our approach allows for 4D imaging of dynamics that cannot be accurately triggered or started, or to dynamics in systems that are never in a static state, such as gels. Instead of imposing additional constraints on the sample reconstruction, NCT is based on the assumption that the dynamic process acting upon the sample can be well described as an arbitrary deformation function that is smooth both temporally and spatially.

## Results

**Nonrigid geometry computed tomography**. In the simplest case, the inconsistency of the tomographic model can be sufficiently described by translation of the sample during data acquisition. This can be caused for example by thermal drifts or imperfection of the nanopositioning system. In this case, the mutual consistency of the projections can be increased by iterative refinement of the projection geometry[13–15] or directly by shifting the measured projections[15–17] in order to minimize the error between the measured data and the tomographic projections.

Estimation and optimal correction of a nonrigid sample deformation is a more complex task with two main challenges. First, since the dynamic process is generally not known a priori, the evolution needs to be recovered from the measurements. The second challenge is an optimal use of the reconstructed deformation field to minimize the amount of required additional information, i.e., to avoid the need of additional projections or added constraints on the sample. Since an improved estimate of the deformation evolution leads to a modified sample reconstruction and vice versa, the reconstruction of a nonrigid sample can

be seen as a joint optimization problem that is generally nonlinear, nonconvex, and needs to be solved iteratively.

If the nonrigid deformation processes are sufficiently slow, periodic or even externally controlled, then it is possible to collect a reference tomogram[9,18], partial tomograms during which the sample is assumed to stay static[2,3,10], or exploit the periodicity of the deformation process to improve the reconstruction[3,4,19]. However, these assumptions can often be too limiting if the deformation process is fast with respect to the tomogram acquisition time or if the process is unrepeatable.

Assuming that the dynamic process can be described by a diffeomorphic deformation, the coarsest deformation model is an affine transformation[1,18,20–22]. The affine transformation provides many advantages, such as exact reconstruction methods[20,23] and direct estimation of the deformation field from the measured projections[18]. However, affine transformations and other methods based on straight-ray projections[22,23] are not general, and in some cases can be an inadequate approximation to describe a realistic deformation processes.

In order to alleviate these limitations, our method is based on the concept of deformation vector fields (DVF)[1]. The time-evolving DVF can more accurately describe the local deformation of the sample features and thus provide a flexible model that allows for a locally and temporally varying deformation. Various DVF-based methods[1,10,12,22] were introduced in the last years for X-ray CT imaging. Here, we extend this concept to samples that are nonlinearly and rapidly evolving with respect to the acquisition rate using multiple partial data sets to provide quality comparable with a motionless sample.

In most experiments, the evolution of the deformation process is unknown and the optimal DVF needs to be estimated to match the measured projections. In order to fully characterize a single time point of the DVF, acquisition of projections from the full angular range, i.e., a half-turn rotation, is preferable. One option to gain this information is the acquisition of projections in an interleaving angular scan protocol, for which the full tomographic scan is split into several subsets of similar number of projections, each containing every n-th angle of the full scan[24]. Here, we refer to such a sub-unit of the data set as a sub-tomogram. It has been already demonstrated that such acquisition schemes work for continuous rotation and help capturing sample time evolution[25].

Given these sub-tomograms, the DVF can be estimated directly from comparison of the partial reconstruction adjacent in time[10–12]. Exact validity of this approach is limited only to samples that are static during acquisition of each sub-tomogram as presented in ref. [10], in which the sample was only deformed between tomogram acquisitions. For samples that deform continuously, but with motion of limited complexity and amplitude, e.g., experiments presented in Refs. [11,12], the latter approach can still provide a good approximation of the DVF and the sample reconstruction. However, in more general cases, in which both the position and the structure of the sample are nonlinearly evolving during the acquisition, the DVF and sample reconstruction should be solved as a joint optimization problem, with an approach that explicitly accounts for changes in the DVF during acquisition.

This work presents an approach, where the DVF evolution is iteratively estimated along with mutual displacements of the measured projections and the sample reconstruction itself. In other words, the sample reconstruction is updated in each iteration given the information about projection displacement and the DVF estimation from the previous iteration. This bootstrapping iterative approach enables convergence to a consistent solution satisfying all measured projections. This approach also enables compensation for the sample deformation on multiple timescales. The DVF estimation method can account for rather slow changes on timescale of a single sub-tomogram.

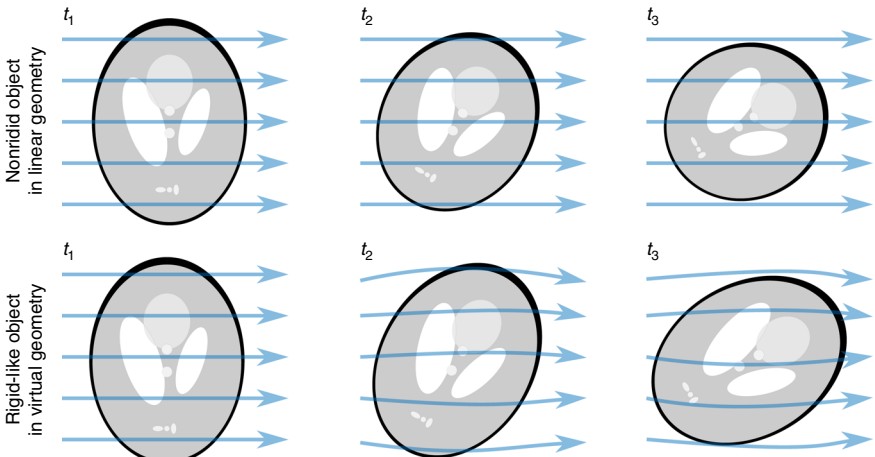

**Fig. 1** Representation of sample deformation via curved-lines projections. (top) Illustration of the parallel geometry with straight lines of sight for a deformed sample compared to equivalent description by a virtual geometry with curved lines of sight for the original, i.e., undeformed, sample (bottom)

On the other hand, mutual displacement can be estimated independently for each acquired projection and thus e.g., rigid motion of the sample can be recovered with much higher time resolution.

In order to avoid reconstruction quality deterioration caused by additional interpolation steps in the reconstruction[10,11], our nonrigid tomography approach was implemented as a transformation of the original straight lines of sight into generally shaped curves as shown in Fig. 1. This leads to a new tomographic geometry that describes the projection measurements by a single rigid volume instead of a time-evolving sequence. The derivation details of our method are described in the Methods section.

One example of a complex deformation process is radiation damage, the changes sustained by the sample when exposed to ionizing radiation, which is particularly relevant for X-ray nanotomography due to the high X-ray dose needed for high-resolution imaging[26–28]. Severe radiation damage can ultimately destroy the imaged features, but already a radiation dose significantly below the maximum tolerable dose leads to deterioration of the reconstruction quality. In order to distinguish these two cases, we will be using the term radiation-induced change (RIC) for the latter case. RIC in a 3D structure can be approximated to first order as a nonrigid deformation process that does not depend on the time of the scan but rather on the total deposited X-ray dose. This means that the problem cannot be circumvented by faster scanning, since certain X-ray dose is always needed to reach the targeted resolution[26,27,29]. RIC is often neglected at the micro- and mesoscale, but it is a severe limitation for X-ray imaging at the nanoscale, since the X-ray dose required to image the sample is inverse proportional to the fourth power of the aimed resolution[26]. This is one of the reasons, why dedicated X-ray nano-CT setups with cryogenic sample protection, which partially mitigates radiation damage effects, are being developed[30–34]. However, cryo nano-CT instruments are not yet common[30], and as discussed in the next section, even cryogenically protected biological samples can still exhibit mild RIC leading to deterioration of the reconstruction quality.

**Numerical simulation**. The reconstruction quality of our NCT method is first demonstrated on an artificial data set, i.e., the phantom. The phantom was modeled as a pillar from a porous material with dimensions of $200 \times 200 \times 100$ pixels shown in Fig. 2a. During the virtual acquisition, the phantom was continuously deformed by a deformation field with amplitude proportional to $1 - \exp(-3t)$, where $t$ is the normalized time

between 0 and 1. The DVF was simulated as a smooth random field with maximal displacement of 10 pixels and characteristic spatial period of 20 pixels. A cut through the DVF model and its time evolution can be seen in Fig. 2e, f, respectively.

From this model, we generated 320 noiseless projections at equidistant angles from 0 to 180 degrees. The virtual acquisition followed an interleaving scanning protocol[24] resulting in four sub-tomograms, each containing every fourth angle. In the following, we refer to these sets of projections that cover the whole 180 degree rotation span, but with larger angular step as sub-tomograms. The volume and DVF reconstruction was performed by 50 iterations of the joint optimization method described in the Methods section. We have used an NCT-based filtered back projection (FBP), described in the Methods section, to reconstruct the sub-tomograms $\mathbf{g}^{(i)}$ and the full tomogram $\mathbf{g}^{(F)}$. Once convergence of the reconstructed DVF was reached, the final reconstruction was further refined by 50 iterations of the NCT-based SIRT (Simultaneous Iterative Reconstruction Technique) method. The DVF estimation by the three-dimensional optical flow method was regularized by an isotropic Gaussian kernel with a standard deviation of 30 pixels.

A tomographic cut through a reconstruction of the data set by a SIRT method based on a static geometry is shown in Fig. 2b, while the reconstruction of the same data set by the NCT-based SIRT method with the self-consistently estimated DVF is shown in Fig. 2c. The original and the reconstructed four-dimensional DVFs are illustrated by the principal component analysis (PCA). Horizontal cuts through their first-principal component are shown in Fig. 2d, e and their time-evolving weights are shown in Fig. 2f. The reconstructed DVF corresponds well to the original DVF model shown in Fig. 2e with RMS error of 0.8 pixel, also its time evolution in Fig. 2f follows closely the model curve. Each of the five interpolation nodes corresponds to the beginning or end of each sub-tomogram with a linear interpolation between the nodes.

The reconstruction quality was additionally quantified by two methods: Fourier shell correlation (FSC)[35] and by the reconstructions gray-scale histograms. The intersection of the FSC curve with the ½-bit threshold curve[35] was used to estimate the average spatial resolution with respect to the known phantom. The FSC between the phantom, depicted in Fig. 2a, and the reconstructions by the conventional and NCT-based methods shown in Fig. 2b, c indicates a significant increase of similarity over all spatial frequencies between the NCT-based SIRT and the phantom compared with the conventional SIRT method. Similarly, the histogram in Fig. 3b shows a more binary-like distribution for the

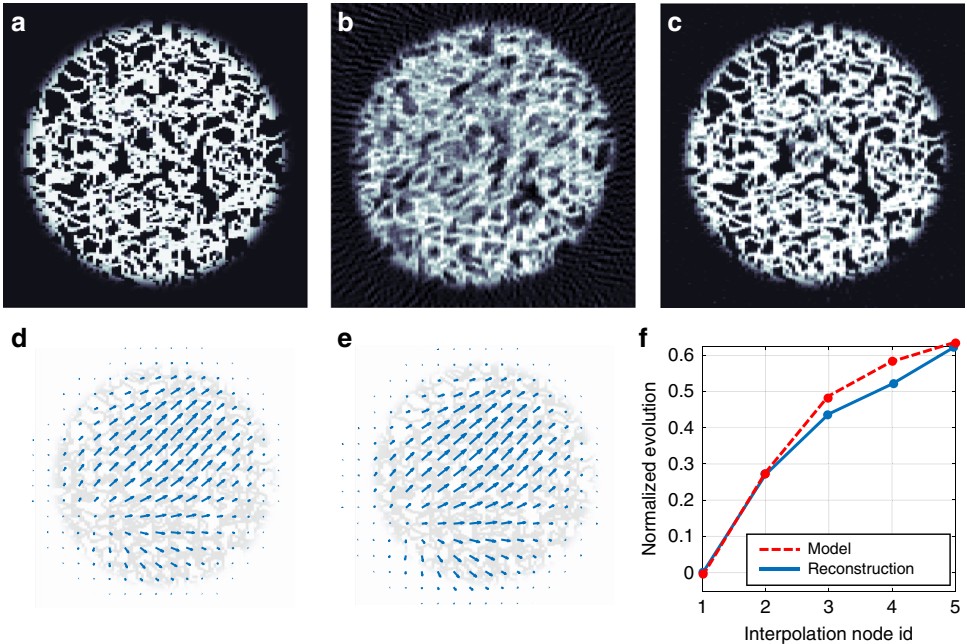

**Fig. 2** Reconstruction quality for a simulated porous material data set. **a** Original phantom, **b** reconstruction of the deformed data set by a standard SIRT method, **c** reconstruction by an NCT-based SIRT method with self-consistently estimated DVF. An axial tomographic cut through the first-principal component of the reconstructed 4D-DVF (**d**) differs from the model DVF (**e**) only by an RMS of 0.8 pixel. The corresponding weights of the principal components, i.e., their time evolution, in the five interpolation nodes are shown in (**f**)

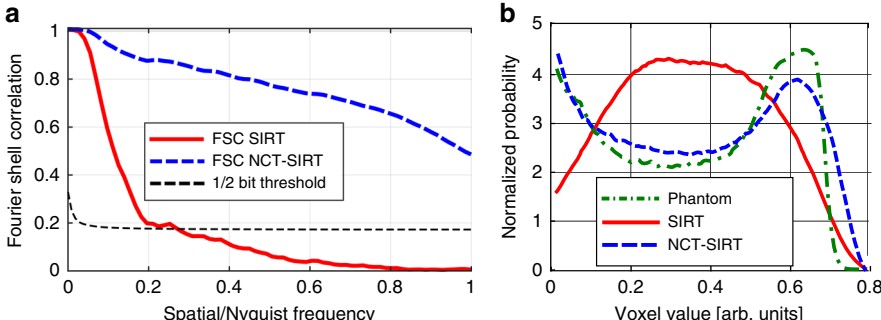

**Fig. 3** Quantification of reconstruction quality improvement. **a** Fourier shell correlation between the original phantom, which is shown in Fig. 2a, and the full reconstructions obtained by the conventional SIRT reconstruction method (red line), shown in Fig. 2b and the NCT-based SIRT reconstruction (blue line) shown in Fig. 2c. **b** Comparison of histograms of the original phantom and both the reconstructions

NCT-based reconstruction method that approximates better that of the original phantom.

**Reconstruction of a biological sample**. The flexibility of the NCT-based reconstruction methods with nonlinear time evolution can be used to improve the reconstruction quality for a more general class of sample changes. Here, we demonstrate the potential of the NCT method for nanoscale tomographic imaging of biological samples obtained by ptychographic X-ray computed tomography (PXCT)[36–38]. Ptychography is a scanning-based coherent diffractive imaging technique that provides high reconstruction robustness and the ability to relax many experimental constraints[39,40]. PXCT method provides high-dose efficiency[41], since no imaging optics is required after the sample, and due to its ability to exploit phase contrast, which at high photon energies provides stronger signal than the absorption contrast.

The imaged biological samples were extracted from scales of *Cyphochilus* beetle, an evolutionary optimized biophotonic material[42]. Two similar samples were imaged, one at room temperature and atmospheric pressure and the second

cryogenically protected during the entire PXCT scan. The non-cryo-protected sample was measured by PXCT with an exposure of 200 ms per position and a step length of 0.3 μm, resulting in a total deposited X-ray dose of $3.8 \times 10^8$ Gy. The angular projections were acquired using an interlaced-angle protocol, resulting in three equally sized sub-tomograms with overall 380 projections. The cryogenically protected sample was imaged with shorter exposures of 100 ms and larger scan step of 0.5 μm in order to reduce the radiation dose to $0.9 \times 10^8$ Gy on a total of 500 angular projections distributed in two sub-tomograms, The data of the cryogenically protected sample are presented in ref. [42]. and permitted detailed analysis of the interaction of this nanophotonic structure with light. In both measurements, the complex-valued projections of the samples were reconstructed by a combination of the difference map[43] and maximum-likelihood[39] algorithms to a pixel size of 14.2 nm.

We split the new tomographic reconstruction procedure into the following steps iteratively executed until convergence:

(1) An initial estimate for the full tomogram, $\mathbf{g}^{(F)}$, is reconstructed from all phase projections, after unwrapping and

aligning, using the NCT-based FBP method initialized with zero deformation, i.e., the conventional FBP method.

(2) The estimates of the 4D-DVF, $\Gamma(\mathbf{x},t)$, are updated using the optical flow method in Eq. (4)

(3) Relative shifts of the unwrapped projections are refined by a gradient-descent method[44] in order to account for the updated DVF $\Gamma(\mathbf{x}, t)$.

The third step is important since relative shifts of the projections $\Psi_\Theta$ are free parameters of ptychography and also the instrumental stability of the PXCT scanners is not sufficient to guarantee sub-30 nm precision for the entire data set, projections for high-resolution PXCT are typically aligned in a postprocessing step. The DVF optimization was performed on a data set that was $4 \times 4$ binned. This leads to faster and more robust DVF estimation assuming that the DVF is sufficiently smooth. Once convergence has been reached, the reconstructed time-evolving DVF was upscaled and used for the final full-resolution reconstruction. We used 50 iterations of the described joint optimization approach, regularized by an isotropic Gaussian kernel with standard deviation of 30 pixels, as described in the Methods section. The final full-resolution reconstruction was refined by 50 iterations of the NCT-based SIRT method without updating the DVF.

Reconstructions of the non-cryogenically protected sample obtained by the standard and the NCT-based SIRT methods are compared in Fig. 4. The radiation induced changes in the sample result in significant smearing artifacts when the sample is reconstructed using conventional tomography, shown in Fig. 4a, c, while the smearing is mostly mitigated in the NCT-based reconstruction in Fig. 4b, d. Note that the standard reconstruction method in Fig. 4a, c provides relatively sharp reconstruction of the center because the self-consistent pre-alignment of the tomographic projections[17] suppresses the rigid-motion artifacts.

The reconstruction quality was further quantified by the Fourier shell correlation curves shown in Fig. 4f and electron-density histograms in Fig. 4e. The FSC-based resolution estimated using the 1/2-bit criterion[35] shows an improvement of resolution from 53 nm to 27 nm. The reduction of the smearing artifacts with the NCT-based SIRT method results in a more binary-like histogram, as expected for this one-phase sample. The histogram of the cryo-protected sample has broader peaks due to lower signal-to-noise ratio in that measurement.

The first PCA mode of the reconstructed DVF, depicted in Fig. 5a in the horizontal and in Fig. 5b in the vertical plane, describes 92% of the total deformation. The reconstructed DVFs illustrate that the radiation-induced deformation can be neither well approximated by a simple affine transform nor by a linear evolution.

Finally, Fig. 6 shows relative shifts of the measured projections which are an integral part of the nanotomography reconstruction procedure[17,44], where the second row shows additional offsets with amplitude up to four pixels that were needed to correct for the imperfect initial guess based on the static tomography model.

Estimation of the DVF for the cryo-protected sample, shown in Fig. 7b, clearly demonstrates that the RIC in the cryo-protected sample with average deformation amplitude of 15.4 nm are significantly lower than under ambient conditions with average deformation amplitude 61 nm. The FSC estimated resolution in Fig. 7a was improved from 33 to 30 nm.

## Discussion

Although the NCT was demonstrated on radiation damage induced changes, applications of our method can be much broader. Since our method is able to account for dynamic sample deformation during each angular sub-tomogram, it could

relax the stringent requirements on the acquisition speed for in-vivo imaging and thus allow to use laboratory-based X-ray phase tomography methods[45,46]. In addition, the nonrigid tomographic geometry is able to better describe complex deformation fields that can originate from internal sample changes[11,18,47,48] and that may not be well described by affine transformations or linear deformation evolution.

We demonstrate a way to improve resolution for imaging radiation-sensitive specimens with non-cryo instrumentation, which are much more widely available in synchrotrons around the world.

Second, but perhaps more importantly, radiation damage is the ultimate limit to the resolution and quality of imaging that can be achieved for any given sample[42], as even samples considered to be radiation hard are reported to suffer from RIC[36,37] when aiming for sub-20-nm resolution. In our work, we provide a path to push this limit further by computationally compensating for the first-order deformation that occurs in the sample.

The improved robustness of the NCT-based tomography methods is gaining even higher importance with the advent of the fourth-generation synchrotron sources that promise more than two orders of magnitude higher coherent X-ray flux[49,50]. In that case, the radiation dose will become the major bottleneck in reaching the full potential of the additional flux for nanoimaging. Our approach will thus result in increasing of the reachable spatial resolution. This is possible because the reconstruction of the initial sample state, $\mathbf{g}^{(F)}$, incorporates information from the whole dynamic data set acquired during the deformation.

Finally, tomography in the curved lines-of-sight geometry can be implemented using graphical processing units (GPUs) in a computationally efficient way[51]. This fast implementation enabled us to use the NCT approach for reconstruction of general samples with 3D deformation field using iterative methods, such as NCT-SIRT.

## Methods

**Tomography in curved geometry**. A tomography reconstruction can be seen as a numerical task that optimizes values of volume $\mathbf{g}$ so that the constraints given by the measured data are well satisfied. For $M$ measurements and $N$ reconstruction pixels, this leads to a system of $M$ linear equations[52]

$$p_j = \sum_1^N a_{ij}g_i + \epsilon_j \text{ for } j = 1, \dots, M \tag{1}$$

where $p_j$ denotes the measured projection data, $a_{ij}$ are contributions of the $i$th voxel of the reconstructed volume $g_i$ to the $j$-th measurement $p_j$ forms that form a sparse matrix $\mathbf{A} : \mathbb{R}^N \to \mathbb{R}^M$ and $\epsilon_j$ is the associated noise. The conventional algebraic tomography reconstruction methods solve the system of Eq. (1) iteratively so that a norm of a difference between the measured data and the reconstructed projections is minimized

$$\min_g \|\mathbf{A}\mathbf{g} - \mathbf{p}\| \tag{2}$$

If the norm is quadratic, the gradient of the cost function with respect to the reconstructed volume $\mathbf{g}$ can be expressed as $\mathbf{A}^T(\mathbf{A}\mathbf{g} - \mathbf{p})$ and the task can be solved by gradient-based solvers.

However, if the object $\mathbf{g}$ is being deformed during the acquisition, the original straight lines of sight cannot describe well the measured data by a unique solution. Instead, the NCT method assumes that the sample changes can be well approximated by an elastic deformation with a smooth time evolution given by a DVF $\Gamma(\mathbf{x}, t)$. $\Gamma(\mathbf{x}, t)$ is defined as a bijective mapping $\mathbb{R}^3 \to \mathbb{R}^3$ from each time frame $t$ of the object $\mathbf{g}(\mathbf{x}, t)$ to a reference time $t_0$. In that case, each line of sight (LoS) $\hat{l}_j$ can be associated with a virtual curved path $\hat{\mathbf{l}}_j$ through a single shared object $\mathbf{g}^{(F)}$, as shown in Fig. 1, resulting in a new non-Euclidean acquisition geometry described by a sparse matrix $\mathbf{A}_N$,

$$\mathbf{A}_N(\Gamma(\mathbf{x}, t))\mathbf{g} = \sum \hat{a}_{ij}g_i \tag{3a}$$

where $\hat{a}_{ij}$ are contributions of the voxels $g_i$ along the virtual path $\hat{\mathbf{l}}_j = \mathbf{l}_j + \Gamma\left(\mathbf{l}_j, t\right)$. The transposed operator $\mathbf{A}_N^T$ that is needed to calculate a gradient-based update could be either directly calculated as transposition of the virtual geometry matrix $\mathbf{A}_N$, however, since the geometry matrix cannot be usually directly expressed due to its

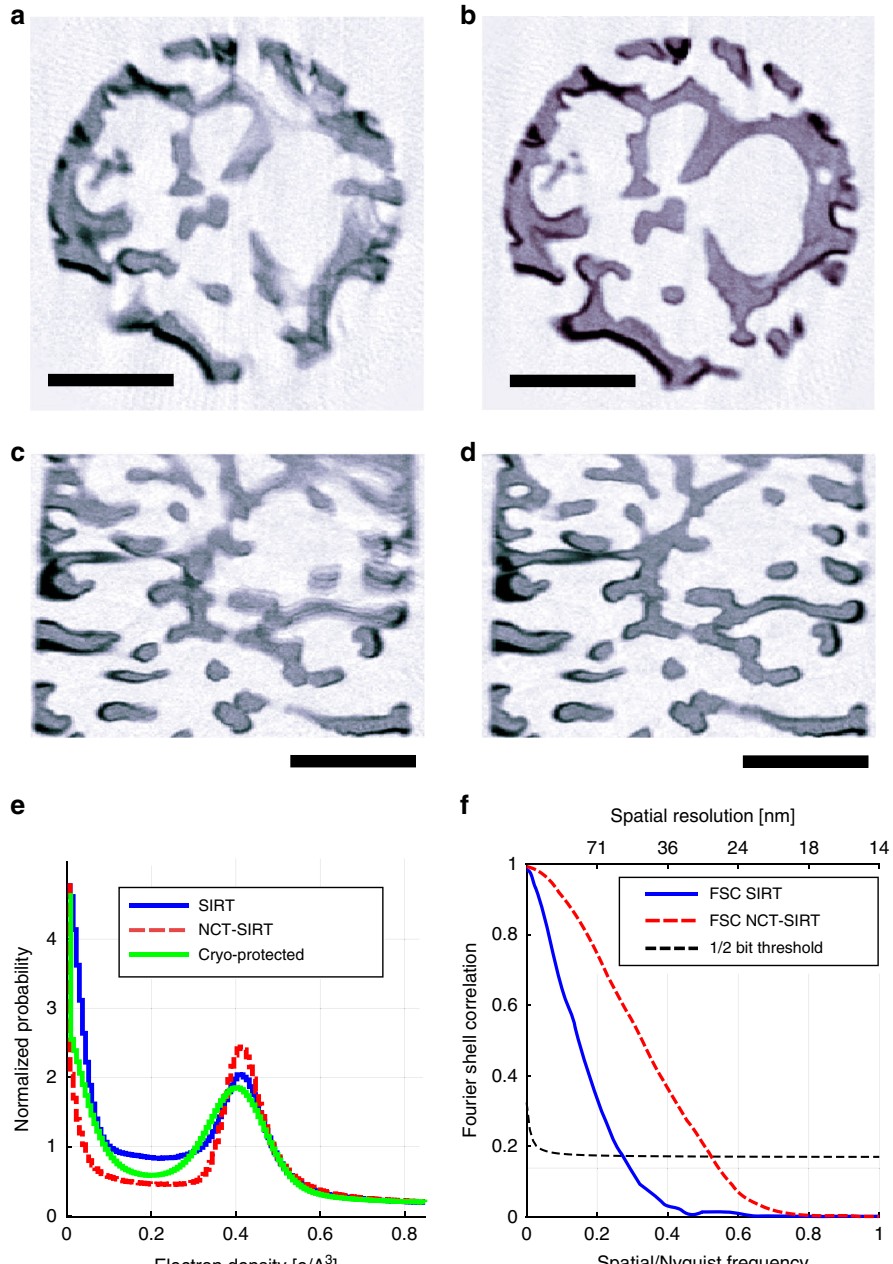

**Fig. 4** Reconstruction of the beetle scale sample imaged at ambient conditions. **a**, **c** shows a horizontal and vertical cut of a common SIRT reconstruction, and **b**, **d** are identical cuts through the NCT-based SIRT method. **e** Histograms of the reconstructed electron density for reconstructions of the ambient and cryogenically protected sample. The reconstruction quality was further quantified by Fourier ring correlation (**f**) showing improvement from 53- nm to 27 -nm resolution. Scale bars denote 2 µm

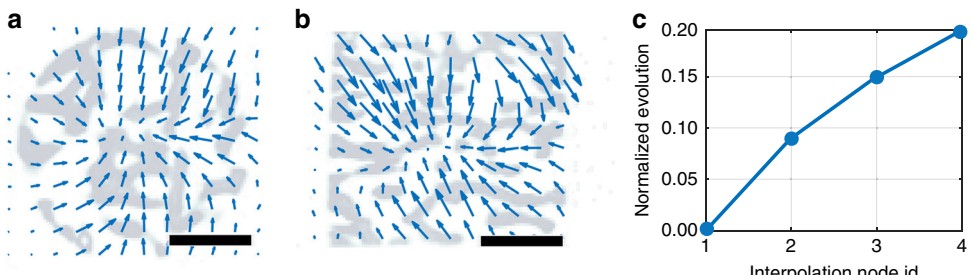

**Fig. 5** Recovered time-evolving DVF for the beetle scale sample. Axial (**a**) and coronal (**b**) cuts through the first-principal component of the reconstructed 4D-DVF and its evolution (**c**) used for the NCT-based reconstruction shown in Fig. 4b, d. Arrows indicating the DVF in **a**, **b** are upscaled 5 × to improve visibility. Scale bars denote 2 µm

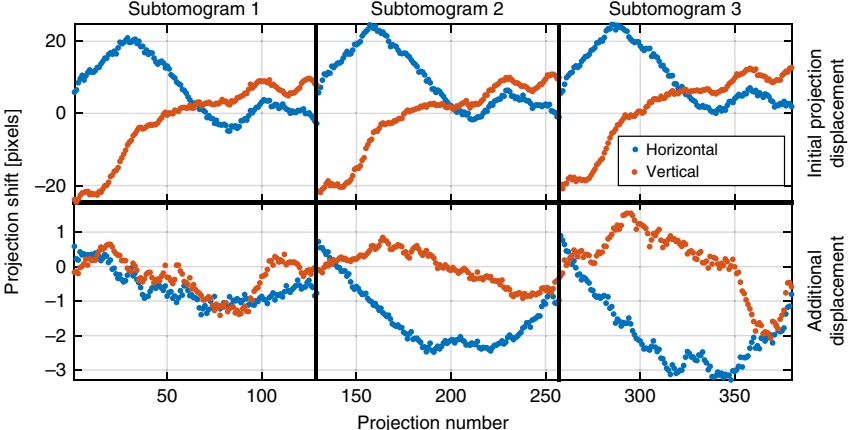

**Fig. 6** Recovered relative shifts of the measured projections. The top row shows initial estimates of the projections shifts for each sub-tomogram. The bottom row shows an additional shift correction that was needed when the nonrigid model of the sample was used for alignment

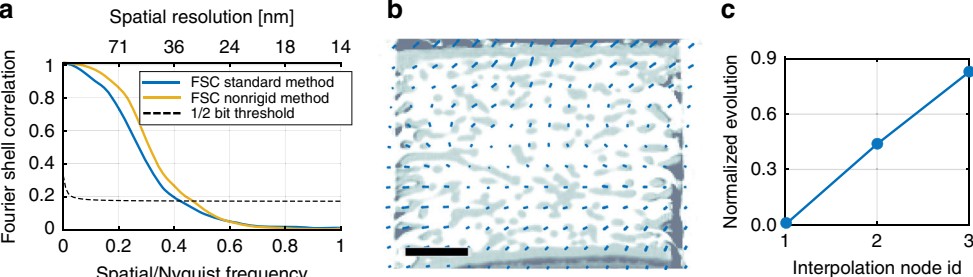

**Fig. 7** A reconstruction quality overview for the cryo-protected beetle sample. **a** Estimation of the resolution of the cryogenically protected sample. **b** The coronal plane view of the reconstructed DVF and its time evolution (**c**) during both sub-tomograms. The plotted DVF vectors were upscaled 5 × to improve visibility. Scale bar denotes 2 μm

size, an alternative option is to express it as,

$$\mathbf{A}_N^{\mathrm{T}}(\Gamma(\mathbf{x}, t))\mathbf{p} = \sum_j \hat{a}_{ji}^{\mathrm{T}} p_j \tag{3b}$$

where $\hat{a}_{ji}^{\mathrm{T}}$ denotes contribution of the $j$-th measured pixel to the $i$th voxel along the virtual path $\hat{\mathbf{l}}_j = \mathbf{l}_j + \Gamma^{-1}(\mathbf{l}_j, t)$. Assuming that the DVF and its inversion are locally and temporally sufficiently smooth, the inverse DVF $\Gamma^{-1}(\mathbf{l}_j, t)$ can be approximated either as $-\Gamma(\mathbf{l}_j, t)$, or more precisely via a simple fixed-iteration scheme[53].

An advantage of the virtual geometry approach is that the optimization task in Eq. (3a) is formally identical to the classical straight-LoS tomography and thus any common algebraic method such as SIRT, SART, or CLGS can be used as long as changes in the sample density resulting from the deformation process can be neglected. This also means our approach can be easily combined with common regularization methods, such as total variation[54].

Because the curved geometry can generally differ for each projection, the NCT-based methods enable reconstruction of samples evolving faster than the acquisition time of a single sub-tomogram if the corresponding DVF evolution is available.

**Estimation of deformation vector fields.** Since the improved estimate of DVF $\Gamma(\mathbf{x}, t)$ affects both the reconstruction $\mathbf{g}^{(\mathrm{F})}$ and also the partial reconstructions $\mathbf{g}^{(i)}$ of each sub-tomogram, reconstruction of the volume $\mathbf{g}^{(\mathrm{F})}$ and DVF $\Gamma(\mathbf{x}, t)$ is generally nonlinear and nonconvex. This means that the estimation of the optimal DVF leads to iterative joint optimization problem when the DVF and the volume are simultaneously reconstructed in order to satisfy all provided constraints.

The continuous time-evolving DVF $\Gamma(\mathbf{x}, t)$ is calculated from the discretized deformation vector fields $\Gamma^{(i)}(\mathbf{x})$, which describe an average deformation in the $i$th sub-tomogram with respect to the reference state. The discretized DVF $\Gamma^{(i)}(\mathbf{x})$ is estimated so that differences between the full reconstruction $\mathbf{g}^{(\mathrm{F})}$ and the reconstructed sub-tomograms $\mathbf{g}^{(i)}$ are minimized. Both $\mathbf{g}^{(i)}$ and $\mathbf{g}^{(\mathrm{F})}$ are already reconstructed using the previous estimate of the time-evolving DVF, $\Gamma(\mathbf{x}, t)$.

Given the current estimate of the full tomogram $\mathbf{g}^{(\mathrm{F})}$ and a sub-tomogram $\mathbf{g}^{(i)}$, a gradient-descent update of the $j$-th axis component of the DVF, $\Gamma_j^{(i)}$, in the $i$th sub-tomogram can be estimated using an in-house implementation of the three-

dimensional optical-flow method[55] with smoothing weights in the following form

$$\tilde{\Gamma}_j^{(i)} = \Gamma_j^{(i)} + \lambda \frac{\left[ \left( \mathbf{g}^{(\mathrm{F})} - \mathbf{g}^{(i)} \right) \nabla_j \mathbf{g}^{(\mathrm{F})} \right] * \mathbf{k}}{\left( \nabla_j \mathbf{g}^{(\mathrm{F})} \right)^2 * \mathbf{k} + \alpha} \tag{4}$$

where $0 < \lambda < 2$ is a relaxation constant, $\mathbf{k}$ denotes a positive convolution kernel, $\nabla_j \mathbf{g}$ is the spatial gradient of the full reconstruction $\mathbf{g}^{(\mathrm{F})}$ along the $j$-th axis, and $\alpha$ is a small constant to avoid amplification of noise in very smooth regions of the reconstruction. The updated DVF can be directly used to refine the estimate of the tomograms. However, due to the block discretisation of the sub-tomograms, this will unavoidably lead to temporal smearing over the duration of a single sub-tomogram, resulting in underestimation of the actual deformation, and to undesired discontinuities between each of the DVF blocks. Therefore, the time evolution of the DVF $\Gamma(\mathbf{x}, t)$ was deconvolved by the Tikhonov method[56] with a regularization term $O(\Gamma(\mathbf{x}, t))$ enforcing smoothness, resulting in the following optimization task:

$$\|\mathbf{S}\,\Gamma(\mathbf{x}, t) - \Gamma^{(i)}(\mathbf{x})\| + \lambda O(\Gamma(\mathbf{x}, t)) \tag{5}$$

where the first term enforces minimal distance between the $i$th discretized DVF $\Gamma^{(i)}$ and the deconvolved DVF $\Gamma(\mathbf{x}, t)$ is averaged over the $i$th sub-tomogram by the block-diagonal sparse matrix $S$. Since estimation of deconvolved $\Gamma(\mathbf{x}, t)$ is generally an ill-posed task, the smoothing functional $O(\Gamma(\mathbf{x}, t))$ aids the reconstruction by providing regularization. In order to avoid the need of calculating $\Gamma(\mathbf{x}, t)$ for every tomographic projection, $\Gamma(\mathbf{x}, t)$ is sampled only at the beginning and the end of each sub-tomogram and linearly interpolated in between.

The simultaneous reconstruction of the DVF and tomogram unavoidably results in additional degrees of freedom, for example scaling and rigid shift of the tomogram in the used parallel tomography geometry. Therefore, we define time $t = 0$ as beginning of the deformation, i.e., $\Gamma(\mathbf{x}, 0) = 0$ and use it as a boundary condition. This effectively means that the reconstructed volume $\mathbf{g}^{(\mathrm{F})}$ will be close to a reconstruction at $t = 0$.

We have observed that for a reasonably small deformation, the convergence of this alternating optimization is rather fast and is usually reached in tens of iterations. The presented algorithms were implemented in Matlab using the parallel computing toolbox for graphics-processing unit (GPU) calculations and a modified ASTRA toolkit[13,57] for calculations of the standard and NCT-based projectors.

**Projection alignment**. The measured projections were aligned simultaneously with the DVF estimation. We have used a projection-matching method, which estimates vertical and horizontal shifts of the measured projection with respect to the projections of the reconstructed volume using the known tomography model. The displacements between these reprojections and the measured projections were estimated by the 2D optical flow method and iteratively corrected for by shifting the measured projections.

**Sample preparation**. A single beetle wing scale was coated with a gold layer of 150 nm. The beetle scales were milled into circular rods with diameter of 10 μm with a focused ion beam (FIB) milling (FEI Philips Dualbeam Quanta 3D), and attached on nanotomography pin-holders[58].

**Experiments**. The PXCT scans were performed using 6.2 keV photon energy coherent X-ray beam at the cSAXS beamline, Paul Scherrer Institut, Switzerland. The samples were scanned across an X-ray probe with a diameter of 3 μm. The sample imaged at ambient conditions was measured in the flOMNI—flexible tOMography Nano Imaging end-station[37]. The second sample was imaged using the OMNY—A tOMography Nano crYo end-station[30] and it was kept at temperature 90 K during entire measurement.

## Data availability

The measured data sets in this study are available in a public repository (https://doi.org/10.5281/zenodo.2578796)[51].

## Code availability

The developed algorithms are available in a public repository (https://doi.org/10.5281/zenodo.2578796)[51].

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

## Acknowledgements

Measurements were carried out at the cSAXS beamline, Paul Scherrer Institut, Switzerland. We thank Ana Diaz for help with PXCT measurements and Ullrich Steiner for fruitful discussions. A.S. acknowledges support from the Shanghai Synchrotron Radiation Facility, Zhangjiang Laboratory, through the Big Data Science Center project. A.S. and X.S. gratefully acknowledge the Adolphe Merkle Foundation for financial support. X.S. acknowledges the support from the Winton Programme for the Physics of Sustainability.

## Author contributions

M.O. and M.G.S. conceived the research project. A.S., X.S. and S.V. prepared the samples. M.H., J.R., A.S., X.S., S.V., C.G.S. and M.G.S. performed the PXCT experiments. M.O. developed the reconstruction code and processed the data. M.O. and M.G.S. wrote the paper.

## Additional information

**Competing interests:** The authors declare no competing interests.

**Journal Peer Review Information:** *Nature Communications* thanks the anonymous reviewers for their contribution to the peer review of this work. Peer reviewer reports are available.

