## [Peer Review File · Nature Communications]

Reviewers' comments:

Reviewer #2 (Remarks to the Author):

This paper presents a method to correct for undesired object motion during X-ray nanotomography, particularly due to radiation damage.

The paper is very well written and structured. The technical details are mentioned clearly and comprehensively. The results are impressive, both for simulated data and real data.

Nevertheless, I have a few remarks:

1/ It is mentioned the method is widely applicable for related techniques. However, its main improvement upon the state of the art (see also a comment below) is the extension to non-linear models due to the acquisition of sub-tomograms. The latter is only possible in methods where the acquisition time of a single exposure is long, hence the dead time caused by rotation can be neglected. In many cases however, a continuous scanning protocol is used and acquiring sub-tomograms is impossible.

2/ The following reference should be mentioned as well: <https://www.nature.com/articles/s41598-017-06333-6> which I will denote reference 57 in the following

3/ As mentioned correctly in the paper, the approach is similar to references 1,10,20,22 (and 57). These 4 (5) references are conceptually similar, but have different approaches. The comparison with and between these 4 (5) references should be (much) more elaborate, clearly indicating the strengths and weaknesses of each cited approach and the added value of the proposed technique with respect to this prior work.

4/ The authors mention they can cope with non-linear DVFs. This is however only true because of the possibility of acquiring sub-tomograms as mentioned earlier. In the approaches 10,22 and 57, a non-linear DVF is also possible, and are considered to be different time-frames in a process. For a non-intended DVF, the different timeframes can nonetheless be combined to a single high-quality scan, which has been presented in reference 22 for a single rotation scan. Furthermore, due to the limited number of points in time and linear interpolation in the different sub-tomograms presented in this work, this aspect should be less emphasized.

5/ In the phantom data, it is a pity that the authors did not exploit the flexibility of simulations to also make the amplitude-time function (the $1-\exp(-3t)$ function) more random over the 3D volume.

6/ In the results section, the NCT-based FBP is mentioned. However, in the methods section it appears that mostly iterative reconstruction schemes are used. Please clarify.

7/ In graphs, e.g. Fig2(f), the different lines are not clear (on black/white print). Please use e.g. dotted lines, and/or different markers.

8/ Fig. 3b: what is the unit of the voxel values?

9/ The resolution is said to be improved from 53nm to 27nm. In my opinion, this evaluation is too simplistic, as the authors also indicate the reconstruction of the center is relatively sharp and therefore the sharpness is not really homogeneous. I do realize it is almost impossible to find a good quantification of this effect, so adding a remark on this simplification is sufficient.

10/ P. 16: "The optimization is based on iterative matching ...", I believe this paragraph is not so clear due to the repetition of the corrected reconstruction of $g^{(i)}$

11/ P. 16 bottom: why the x-axis?

12/ Was this performed using a fully custom implementation of the DVF estimation, or did you use any available toolkits? Please clarify and discuss in more detail.

13/ The combination with the misalignment correction is not sufficiently elaborated on.

Globally, this is a very interesting manuscript and although the added value as compared with some of the mentioned references is relatively limited in terms of methodology, I believe the combination of the importance of this topic, the application in nanotomography and the prospect of reducing radiation induced deformation in this technique definitely justifies publication in Nature Communications. In this respect, comment number 3 is obviously the most important one and should definitely be taken into account in the revised version of this manuscript.

Reviewer #3 (Remarks to the Author):

The manuscript "Ab initio nonrigid x-ray nanotomography" presents a timely and convincing treatment of dynamic tomography, sometimes also denoted by 4D tomography, i.e. tomography of non-static samples. The particular implementation of the algorithm seems very convincing and general. A major step forward is the fact that the 3d vector flow field is estimated from the data in a consistent and regularized manner, based on reasonable smoothness constraints, and that the deformation model is not limited to affine transformation.

My only concern with regard to the novelty requirements of Nature Communication, is the incomplete delineation with respect to prior art. This has to be significantly improved. For example, Rit, Sarrut and Desbat (IEEE transactions on medical imaging, 28(10):1513{1525, 2009) have proposed a solution built on very similar ideas, not limited to affine transformations (see Fig.3). This has to be cited and discussed early on in the manuscript; similarly the work on 'tomography using dynamically curved path' by Ruhlandt et al, which is cited only in a very unspecific, general manner. The fact that the data of the PXCT has been published in part (cryogenic measurement) elsewhere, and in view of the in my limited 'structural information' gained by the reconstruction shown in Fig.4, this issue of (relative) novelty in the method/algorithm a central question, concerning suitability for Nat. Comm.. The NCT approach presented here based on the smooth time evolution of the DVT seems to go beyond all of the previous work and may open up much further progress. On the other hand, there exist already quite a few approaches which use optical flow methods and/or distorted and curved tomography geometries in order to compensate for motion. The current work has to be better delineated and discussed with respect to this work. For this purpose, it does not matter whether the scope of the tomography is medical or analytical, nor the length scales or whether projections are required in full-field or by PXCT. If offered publication in Nat.Comm. - provided a corresponding revision - I also recommend that the new algorithm and numerical implementation should be published as meta material, in order to test the performance and relative gain in the presented approach over prior art. The algorithm would be much more important here than the actual data set.

Reviewer #2 (Remarks to the Author):

This paper presents a method to correct for undesired object motion during X-ray nanotomography, particularly due to radiation damage.

The paper is very well written and structured. The technical details are mentioned clearly and comprehensively. The results are impressive, both for simulated data and real data.

Nevertheless, I have a few remarks:

1/ It is mentioned the method is widely applicable for related techniques. However, its main improvement upon the state of the art (see also a comment below) is the extension to non-linear models due to the acquisition of sub-tomograms. The latter is only possible in methods where the acquisition time of a single exposure is long, hence the dead time caused by rotation can be neglected. In many cases however, a continuous scanning protocol is used and acquiring sub-tomograms is impossible.

ANSWER: *We thank the reviewer for their positive assessment. We would like to remark that an acquisition of subtomograms with interlaced angles is also possible with continuous angular motion as was demonstrated in Ref 26, in the updated manuscript. With such acquisition schemes already experimentally demonstrated we reinstate the wide applicability of our method. To clarify this point we have added to our manuscript:*

“One option to gain this information is the acquisition of projections in an interleaving angular scan protocol, for which the full tomographic scan is split into several subsets of similar number of projections, each containing every n-th angle of the full scan. Here we refer to such a sub-unit of the dataset as a sub-tomogram. It has been already demonstrated that such acquisition schemes work for continuous rotation and help capturing sample time evolution²⁶.

2/ The following reference should be mentioned as well:

<https://www.nature.com/articles/s41598-017-06333-6> which I will denote reference 57 in the following

ANSWER: *Thank you. This paper was already cited in our manuscript as Ref 47.*

3/ As mentioned correctly in the paper, the approach is similar to references 1,10,20,22 (and 57). These 4 (5) references are conceptually similar, but have different approaches. The comparison with and between these 4 (5) references should be (much) more elaborate, clearly indicating the strengths and weaknesses of each cited approach and the added value of the proposed technique with respect to this prior work.

ANSWER: *Our method is trying to achieve similar goals as the mentioned references 1,10,20,22,47(57), i.e. use flexibility of tomography method to account for sample changes that can be described as a deformation of the sample. The major difference is our effort to search for single self-consistent solution that satisfy all the measured data without adding*

unnecessary degrees of freedom or assumptions. For example Ref 1 uses only a simple linear approximation of the changes caused by the sample deformation. In Ref 10 no motion is assumed within the acquired subtomograms. Ref. 20 works only with fan beam geometry and cannot describe a general 3D deformation. Finally, Ref 22 assumes only 2D deformation and thus it neglects important degrees of freedom to be optimized. Our method allows us to iteratively improve both the reconstruction and the deformation field and thus maximize consistency between the measured projections.

The advantages of our approach with respect to previous work, e.g. the mentioned references 1,10,20,22,47, gets more pronounced for samples with more nonlinear temporal evolution of the deformation field. In contrast to the simple-model experiments of vertical compression with constant velocity used in Refs 10,22. Additionally, the approach of curved lines of sight we implemented avoids repeated interpolation compared to the approach implemented in Refs 10,47(57).

We have attempted to clarify these differences in our manuscript and put our work better in context with this prior work:

“Given these sub-tomograms, the DVF can be estimated directly from comparison of the partial reconstruction adjacent in time^{10,12,13}. However, exact validity of this approach is limited to samples that are static during acquisition of each subtomogram¹⁰ and it is a good approximation for samples that evolve with constant velocity^{12,13}. In the general case, when the position and the structure of the sample are nonlinearly evolving during a sub-tomogram acquisition and its evolution cannot be well approximated by a constant speed model, the optimal DVF and sample reconstruction needs to be solved as a joint optimization problem.”

and:

“In order to avoid reconstruction quality deterioration caused by additional interpolation steps in the reconstruction^{10,12}, our nonrigid tomography approach was implemented as a transformation of the original straight lines of sight into generally shaped curves as shown in Fig 1.”

4/ The authors mention they can cope with non-linear DVFs. This is however only true because of the possibility of acquiring sub-tomograms as mentioned earlier. In the approaches 10,22 and 57, a non-linear DVF is also possible, and are considered to be different time-frames in a process. For a non-intended DVF, the different timeframes can nonetheless be combined to a single high-quality scan, which has been presented in reference 22 for a single rotation scan.

ANSWER: *The main idea and difference compared to the previous literature is proper use of the bootstrap approach to estimate the DVF, i.e. iteratively refine the reconstructed volume and DVF and search for one consistent solution. The advantage of our approach compared to the previously published methods in [10,22,57(47)] is mainly for DVF with nonlinear evolution. If the evolution would be very close to linear, and the DVF evolution rather simple, e.g. metal foam deformed by constant compression rate, the approximation that the DVF is estimated as a motion between two sub-reconstructions can be reasonably well valid. However, if the rate of change is not constant, i.e. deformation accelerates or decelerates*

during acquisition, DVF estimation from comparison of two blocks will become less precise and a proper deconvolution, as implemented in our approach, is needed.

We have modified our text in the following way:

“Given these sub-tomograms, the DVF can be estimated directly from comparison of the partial reconstruction adjacent in time^{10,12,13}. However, exact validity of this approach is limited to samples that are static during acquisition of each subtomogram¹⁰ and it is a good approximation for samples that evolve with constant velocity^{12,13}. In the general case, when the position and the structure of the sample are nonlinearly evolving during a sub-tomogram acquisition and its evolution cannot be well approximated by a constant speed model, the optimal DVF and sample reconstruction needs to be solved as a joint optimization problem.”

Furthermore, due to the limited number of points in time and linear interpolation in the different sub-tomograms presented in this work, this aspect should be less emphasized.

ANSWER: *The limited number of points in time and DVF with uncontrolled and nonlinear evolution is the most difficult combination to reconstruct, so we think putting emphasis there is well justified. If there are many time points available, as e.g. in Ref 57(47), the total object variation over the entire scan may seem to be large, but differences between the subsequent tomograms can be small, making the linear approximation more likely to be valid and simplifying the reconstruction process. In addition, the rate of change, i.e. the amplitude differences in subsequent DVF, will be small. In our example of radiation damage, the deformation rate was approximately $\exp(-K * \text{dose})$. This leads to rather abrupt sample changes during the several first few sub-tomograms. These initial data points, for which the temporal sampling is sparser, are however the most important to recover because the fine sample structure is not yet damaged by radiation and can be still extracted.*

5/ In the phantom data, it is a pity that the authors did not exploit the flexibility of simulations to also make the amplitude-time function (the $1 - \exp(-3t)$ function) more random over the 3D volume.

ANSWER: *Although we had originally considered the option to use a more complex time evolution, we finally have decided to use our time evolution model close to the expected evolution of the radiation-deformed dataset.*

6/ In the results section, the NCT-based FBP is mentioned. However, in the methods section it appears that mostly iterative reconstruction schemes are used. Please clarify.

ANSWER: *FBP was used only for initialization, while the SIRT-based methods are used for the iterative refinement of the full resolution reconstruction. This is explained in our manuscript as follows:*

“We have used an NCT-based filtered backprojection (FBP), described in the Methods section, to reconstruct the sub-tomograms $\mathbf{g}^{(i)}$ and the full tomogram $\mathbf{g}^{(F)}$. Once convergence of the reconstructed DVF was reached, the final reconstruction was further refined by 50

iterations of the NCT-based SIRT (Simultaneous Iterative Reconstruction Technique) method."

We have added clarification to these points, where the selected method may have been ambiguous.

7/ In graphs, e.g. Fig2(f), the different lines are not clear (on black/white print). Please use e.g. dotted lines, and/or different markers.

ANSWER: *Thank you for the remark, the line styles are now clearly separated.*

8/ Fig. 3b: what is the unit of the voxel values?

ANSWER: *In the simulated dataset, the voxel values are in arbitrary units and they have meaning only in comparison with the values in the original phantom. However, we have added [a.u.] unit mark to avoid confusion.*

9/ The resolution is said to be improved from 53nm to 27nm. In my opinion, this evaluation is too simplistic, as the authors also indicate the reconstruction of the center is relatively sharp and therefore the sharpness is not homogeneous. I do realize it is almost impossible to find a good quantification of this effect, so adding a remark on this simplification is sufficient.

ANSWER: *We agree. This is a general problem on how to define a unique number for resolution, because the resolving power of any imaging system can vary with a wide range of variables, such as position, orientation, depth, etc., To clarify this point we have added a remark that the FSC is providing an average value of resolution for the entire reconstruction volume:*

"The intersection of the FSC curve with the 1/2-bit threshold curve³⁵ was used to estimate the average spatial resolution with respect to the known phantom."

10/ P. 16: "The optimization is based on iterative matching ...", I believe this paragraph is not so clear due to the repetition of the corrected reconstruction of $g^{(i)}$

ANSWER: *Thank you for the comment, we have modified the paragraph to make the content more clear*

The continuous time-evolving DVF $\Gamma(t,x)$ is calculated from the discretized deformation vector fields $\Gamma^{(i)}(x)$, which describe an average deformation in the i -th sub-tomogram with respect to the reference state. The discretized DVF $\Gamma^{(i)}(x)$ are estimated so that differences between the full reconstruction $g^{(F)}$ and the reconstructed sub-tomograms $g^{(i)}$ are minimized.

Both $\mathbf{g}^{(i)}$ and $\mathbf{g}^{(F)}$ are already reconstructed using the previous estimate of the time evolving DVF, $\Gamma(\mathbf{x}, t)$.

11/ P. 16 bottom: why the x-axis?

ANSWER: This was only an example for one axis. We have modified the notation to clarify that the calculation is performed along each axis

12/ Was this performed using a fully custom implementation of the DVF estimation, or did you use any available toolkits? Please clarify and discuss in more detail.

ANSWER: We have implemented our custom version of the GPU accelerated 3D-dimensional optical flow method. The reason behind this was that we did not find any ready-made toolkit for GPU that would be well integrated in Matlab and thus could be used for our iterative refinement. GPU implementation is important to avoid CPU/GPU communication during tomography reconstruction and the subsequent DVF refinement, which makes it very fast. We have now clarified this by:

“ the DVF $\Gamma^{(i)}$ in the i -th sub-tomogram can be estimated using an in-house implementation of the three-dimensional optical flow method⁵⁴”

13/ The combination with the misalignment correction is not sufficiently elaborated on.

ANSWER: We have added more details to clarify advantages of simultaneous refinement of DVF and geometry into our manuscript:

“This work presents an approach, where the DVF evolution is iteratively estimated along with mutual displacements of the measured projections and the sample reconstruction itself. This approach enables compensation for the sample deformation on multiple timescales. The DVF estimation method can account for rather slow changes on timescale of a single sub-tomogram. On the other hand, mutual displacement can be estimated independently for each acquired projection and thus e.g. rigid motion of the sample can be recovered with much higher time resolution”

We have also provided a brief description of the alignment in the Methods section:

Projection alignment

The measured projections were aligned simultaneously with the DVF estimation. We have used a projection-matching method, which estimates vertical and horizontal shifts of the measured projection with respect to the projections of the reconstructed volume using the current iterate of the tomography reconstruction. The displacements between these computational projections and the measured projections were estimated by the 2D optical flow method and iteratively corrected for by shifting the measured projections.

Additional details including exact implementation of this alignment will be provided in a separated manuscript that is in preparation.

Globally, this is a very interesting manuscript and although the added value as compared with some of the mentioned references is relatively limited in terms of methodology, I believe the combination of the importance of this topic, the application in nanotomography and the prospect of reducing radiation induced deformation in this technique definitely justifies publication in Nature Communications. In this respect, comment number 3 is obviously the most important one and should definitely be taken into account in the revised version of this manuscript.

ANSWER: *We thank the reviewer again for their positive assessment on the impact of our work and its suitability for Nature Communications.*

Reviewer #3 (Remarks to the Author):

The manuscript "Ab initio nonrigid x-ray nanotomography" presents a timely and convincing treatment of dynamic tomography, sometimes also denoted by 4D tomography, i.e. tomography of non-static samples. The particular implementation of the algorithm seems very convincing and general. A major step forward is the fact that the 3d vector flow field is estimated from the data in a consistent and regularized manner, based on reasonable smoothness constraints, and that the deformation model is not limited to affine transformation.

My only concern with regard to the novelty requirements of Nature Communication, is the incomplete delineation with respect to prior art. This has to be significantly improved. For example, Rit, Sarrut and Desbat (IEEE transactions on medical imaging, 28(10):1513{1525, 2009) have proposed a solution built on very similar ideas, not limited to affine transformations (see Fig.3). This has to be cited and discussed early on in the manuscript; similarly the work on 'tomography using dynamically curved path' by Ruhlandt et al, which is cited only in a very unspecific, general manner. The fact that the data of the PXCT has been published in part (cryogenic measurement) elsewhere, and in view of the in my limited 'structural information' gained by the reconstruction shown in Fig.4, this issue of (relative) novelty in the method/algorithm a central question, concerning suitability for Nat. Comm.. The NCT approach presented here based on the smooth time evolution of the DVT seems to go beyond all of the previous work and may open up much further progress. On the other hand, there exist already quite a few approaches which use optical flow methods and/or distorted and curved tomography geometries in order to compensate for motion. The current work has to be better delineated and discussed with respect to this work. For this purpose, it

does not matter whether the scope of the tomography is medical or analytical, nor the length scales or whether projections are required in full-field or by PXCT. If offered publication in Nat.Comm. - provided a corresponding revision - I also recommend that the new algorithm and numerical implementation should be published as meta material, in order to test the performance and relative gain in the presented approach over prior art. The algorithm would be much more important here than the actual data set.

ANSWER: *We thank the reviewer for their positive remarks and their assessment in regard of the quality of our work and even its potential to “open up much further progress”. Indeed various attempts for implementation of tomography for dynamic or deformed samples were already published over the last decade including both aforementioned articles. The goal in all of them is clear, extract the local information that is present in the measurement and lost only due to invalid assumptions about the sample stability of conventional reconstruction methods.*

Compared to the first publication mentioned by the reviewer, Rit, Sarrut and Desbat (2009), our method allows a much more general class of deformations. The method in Rit, et al. (2009) assumes the deformation process to be decomposable into so-called “shear-warp” deformation. This is a very restricted class of deformations compared to the more general approach presented in our method, in which any spatially smooth deformation can be considered. A “shear-warp” deformation would not be able to describe well a complex deformation processes such as radiation damage. Additionally, as the authors Rit et al. wrote: “the motion of the mechanical phantom was limited to a cranio-caudal translation which belongs to the category of deformations that can be exactly compensated”. This cannot be generally assumed and our NCT method does not rely on such strong assumptions. Finally, Rit et al wrote in their work: “the estimation of patient motion was not precise because it relied on the hypotheses that the motion is regular and identical to the motion of the patient during the acquisition of the 4-D CT image on the conventional CT scanner”. The fundamental contribution of our method is its applicability on aperiodic non-repeatable deformation processes, where it is not possible to acquire more data and improve reconstruction by assuming periodicity of the motion.

We have modified our manuscript as follows:

The affine transformation provides many advantages such as exact reconstruction methods^{21,24} and direct estimation of the deformation field from the measured projections¹⁹. However, affine transformations and other methods based on straight ray projections^{23,24} are not general and in some cases can be an inadequate approximation to describe a realistic deformation processes.

In order to alleviate these limitations, our method is based on the concept of deformation vector fields (DVF)¹. The time evolving DVF can more accurately describe the local deformation of the sample features and thus provide a flexible model that allows for a locally and temporally varying deformation. Various DVF-based methods^{1,10,13,23} were introduced in the last years for X-ray CT imaging. Here, we extend this concept to samples that are nonlinearly and rapidly evolving with respect to the acquisition rate using multiple partial datasets to provide quality comparable with a motionless sample.

The second mentioned reference: Ruhlandt et al (2017), removed some limitations of the work in the previously mentioned Rit, Sarrut and Desbat (2009) by allowing more general deformations. However, their method requires additional interpolation steps, which as discussed e.g. in Rit, Sarrut and Desbat (2009), will lead to reconstruction quality deterioration. Additionally, the deformations recovered in the work Ruhlandt et al. were very small due to large acquisition speed with respect to the deformation evolution and assumptions of their method that the time evolution of the deformation field is close to linear were rather easily satisfied. Finally, their approach is rather impractical due to its high computational cost, which could be one of the reasons why the authors applied the method only on 2D tomography instead of accounting for general 3D deformation or even combining it with computationally more demanding iterative tomography solvers (SIRT/SART). We have observed that both these steps, i.e. full 3D deformation model and SIRT solver, are important to mitigate artefacts, which could be present in the curved tomographic geometry.

We have modified our manuscript to compare our work in more detail with these prior work and put it better in context with the literature:

“Given these sub-tomograms, the DVF can be estimated directly from comparison of the partial reconstruction adjacent in time^{10,12,13}. However, exact validity of this approach is limited to samples that are static during acquisition of each subtomogram¹⁰ and it is a good approximation for samples that evolve with constant velocity^{12,13}. In the general case, when the position and the structure of the sample are nonlinearly evolving during a sub-tomogram acquisition and its evolution cannot be well approximated by a constant speed model, the optimal DVF and sample reconstruction needs to be solved as a joint optimization problem.”

And:

Finally, tomography in the curved lines-of-sight geometry can be implemented using graphical processing units (GPUs) in a computationally efficient way⁵². This fast implementation enabled us to use the NCT approach for reconstruction of general samples with 3D deformation field using iterative methods such as NCT-SIRT.

However, the major advantage of our approach compared to the mentioned articles is the ability to search for a single solution that satisfies all the measured data and allow for a somewhat arbitrary deformation process, all this while providing only the minimal degree of freedom needed to describe the deformation.

Additionally, our approach combines nonrigid tomography with rigid motion corrections, i.e. mutual alignment of the projections, which are critical for nanotomography. We have observed that this combination even further improves robustness of the nonrigid matching and, in contrast to the nonrigid correction, the time resolution of the rigid motion correction is not limited to time span of a single subtomogram. .

Finally regarding the next reviewer’s comment “and in view of the in my limited 'structural information' gained by the reconstruction shown in Fig.4, this issue of (relative) novelty in the method/algorithm a central question”.

We would contend that the difference between Fig.4(a) and 4(b), i.e. before and after the application of our method, is not “limited gain” but actually rather quite dramatic. Our method allows resolving the actual position of features and surface details with an average resolution improvement from 53 nm to 27 nm. As can be seen in Fig. 4(a), in the center of the reconstruction the artifacts are more pronounced, locally in this regions the improvement is likely a factor of 4 or better. Finally, for the purpose of electromagnetic simulations, which were intended for this sample to elucidate the effect of the structure on the incident light, the accurate recovery of the local electron density is crucial, and can be seen clearly improved in these figures.

Although Figure 4 presents a real case scenario were non-repeatable, and non-reversible changes were occurring in the sample, the result can be validated by the cryogenic measurement shown in Figure 7.

In summary, our work clearly demonstrates two advantages. One is the study of radiation sensitive specimens with non-cryo instrumentation, which is much more widely available in synchrotrons around the world. Secondly, but perhaps more importantly, radiation damage is the ultimate limit to the resolution and quality of imaging that can be achieved for any given sample. Our work provides a path to push this limit further by computationally compensating for the first order deformation that occurs in the sample.

In order to emphasize this point we have modified the text as follows:

We demonstrate a way to improve resolution for imaging radiation-sensitive specimens with non-cryo instrumentation, which are much more widely available in synchrotrons around the world.

Secondly, but perhaps more importantly, radiation damage is the ultimate limit to the resolution and quality of imaging that can be achieved for any given sample, as even samples considered to be radiation hard are reported to suffer from RIC^{37,38} when aiming for sub-20nm resolution. In our work, we provide a path to push this limit further by computationally compensating for the first order deformation that occurs in the sample.

Finally to answer the last reviewers comment: I also recommend that the new algorithm and numerical implementation should be published as meta material, in order to test the performance and relative gain in the presented approach over prior art. The algorithm would be much more important here than the actual data set.

ANSWER: *We have decided to make our code publically available upon publication including test datasets.*

Data availability section was modified:

The measured datasets in this study and developed algorithms are available in a public repository (doi:10.5281/zenodo.2578796)⁵².

Reviewers' comments:

Reviewer #2 (Remarks to the Author):

The authors have done great effort in replying to the comments (from both reviewers). Based on these adjustments, I only have a few minor comments and remarks. I will use the same numbering as in the rebuttal.

2) My apologies for overlooking this (yet it confirms the comment of Reviewer #3 on this citation)

3) The additional paragraphs have improved the manuscript significantly. However, from my interpretation of the cited papers, I think there are still a few minor changes required:

- In the rebuttal, the authors claim Ref 22 (current Ref 13) assumes only 2D deformation, which I believe is not true.

- In the first new paragraph, the authors state "[...] that are static during acquisition of each subtomogram(10) and it is a good approximation [...]". This "and" should be replaced by "or", as it is not clear that the first claim is not valid for the two other references.

4) The authors appear to underestimate the complexity of the behaviour of foam under (constant) compression, which is far from linear. Both (current) refs 10 and 13 do cope with these complex motions if sufficient datapoints in time are acquired. This also comes back to my first comment. Ref. 26 (revised manuscript) would in this case mean more time steps can be taken also for the methods in refs 10 and 13, allowing for a more complex motion. In that sense, the claim that "it is a good approximation for samples that evolve with a constant velocity(12,13)" is too strong, and should be altered (e.g. "motion with limited complexity"). In Ref 13, "case 3" represents this scenario, yet it is unclear whether a more complex interpolation scheme is taken into account. As a result, I still believe the authors emphasize too much the interpolation and the complexity of the motion. As far as I can tell from the relatively limited details both in this paper and the cited references (inherent to the multidisciplinary journals), there is however a fundamental improvement that needs more emphasis, i.e. what the authors call the "iterative refinement", partially inherent to the SIRT method, which is in literature only achieved once for every (sub)tomogram, for different reasons: in 10 this is a design choice, in 12 and 13 this is inherent to the reconstruction technique (respectively FBP and SART, where in the latter a single iteration covers the whole subtomogram dataset). Although the processes shown in these three references are definitely not less complex (I would even say the process shown in 12 is more complex than the RIC shown in this paper), it is possible that due to this less frequent update these methods are more prone to sudden events.

As mentioned, I believe this is very nice work and it is a very good paper that deserves publication in Nature Communications, albeit with implementation of some more detailed discussion on the literature (notable 10,12, 13)

Reviewer #3 (Remarks to the Author):

The authors have answered all questions convincingly. I particularly appreciate that they will disclose the code.

I am still sceptical about deducing information from a tomographic reconstruction where severe changes occur due to radiation damage, but this does not stand against using this method for more relevant problems with intrinsic dynamics. I recommend publication as is.

Reviewer #2 (Remarks to the Author):

The authors have done great effort in replying to the comments (from both reviewers). Based on these adjustments, I only have a few minor comments and remarks. I will use the same numbering as in the rebuttal.

2) My apologies for overlooking this (yet it confirms the comment of Reviewer #3 on this citation)

Answer: *Reference to Ruhland et al is now further discussed in the introduction. No further changes were added into our manuscript.*

3) The additional paragraphs have improved the manuscript significantly. However, from my interpretation of the cited papers, I think there are still a few minor changes required:

- In the rebuttal, the authors claim Ref 22 (current Ref 13) assumes only 2D deformation, which I believe is not true.

Answer: *We are very sorry, this was misreference and in our rebuttal we meant 47 instead of 22. Additionally, we now see that our characterization that Ref 47 (now Ref 12 in this version) assumes only 2D deformation was inaccurate. More precisely, we meant in our response that in Ref 47 only 2D deformation of each tomographic slice is considered. In the end this allows some 3D characteristics of the deformation, but it explicitly excludes deformation of features in the direction of the rotation axis and therefore is of somewhat limited application.*

- In the first new paragraph, the authors state "[...] that are static during acquisition of each subtomogram(10) and it is a good approximation [...]". This "and" should be replaced by "or", as it is not clear that the first claim is not valid for the two other references.

Answer: *We have modified this sentence to make clear the connection between the two statements.*

Original text:

However, exact validity of this approach is limited to samples that are static during acquisition of each subtomogram¹⁰ and it is a good approximation for samples that evolve with constant velocity^{12,13}. In the general case, when the position and the structure of the sample are nonlinearly evolving during a sub-tomogram acquisition and its evolution cannot be well approximated by a constant speed model, the optimal DVF and sample reconstruction needs to be solved as a joint optimization problem.

Modified text:

Exact validity of this approach is limited only to samples that are static during acquisition of each subtomogram as presented in Ref 10, in which the sample was only deformed between tomogram acquisitions. For samples that deform continuously, but with motion of limited complexity and amplitude, e.g. experiments presented in Refs. 12,13, the latter approach can still provide a good approximation of the DVF and the sample reconstruction. However, in more general cases, in which both the position and the structure of the sample are

nonlinearly evolving during the acquisition, the DVF and sample reconstruction should be solved as a joint optimization problem, with an approach that explicitly accounts for changes in the DVF during acquisition.

4) The authors appear to underestimate the complexity of the behaviour of foam under (constant) compression, which is far from linear. Both (current) refs 10 and 13 do cope with these complex motions if sufficient datapoints in time are acquired. This also comes back to my first comment. Ref. 26 (revised manuscript) would in this case mean more time steps can be taken also for the methods in refs 10 and 13, allowing for a more complex motion. In that sense, the claim that "it is a good approximation for samples that evolve with a constant velocity(12,13)" is too strong, and should be altered (e.g. "motion with limited complexity"). In Ref 13, "case 3" represents this scenarion, yet it is unclear whether a more complex interpolation scheme is taken into account.

Answer: *We have decided to follow the suggestion of reviewer #2, because the method in Refs 12,13 indeed provides a good approximation for some motions with limited complexity and amplitude, which was well demonstrated in Refs 12 and 13 . Therefore, we have replaced "constant velocity motion" by "motion with limited complexity and amplitude" in our text, as can be read in the previous answer above. As reviewer 2 noted, some of the implementation details of Ref 13 are not very clear, however the applied interpolation approach is well described in Eqs 8 and 9. The authors of Ref 13 explain that they use "a simple linear evolution of a displacement field" between each registration timestamp, i.e. their deformation model describes sample deformation with a DVF that has a linear evolution with time between the timestamps. However, the major difference is in the way, how the DVF is estimated because as authors of Ref 13 noted: "The technique falls or stands with a good model for the displacement field's evolution at each time point t_p ." As can be read in the following answer, we have attempted to emphasize the novelty of our approach in estimating the DVF. Also, as we have described in our previous answer, Ref 10 assumes sample to be quasi static, i.e. the sample is not changing during the partial tomography scans, and the sample is deformed by controlled external forces only in between the scans. This is a significantly different and easier problem than the one we tackle, for which the sample is uncontrollably changing during the tomography measurements. We have also clarified these points further in the modified text shown in the response to the previous point above.*

As a result, I still believe the authors emphasize too much the interpolation and the complexity of the motion. As far as I can tell from the relatively limited details both in this paper and the cited references (inherent to the multidisciplinary journals), there is however a fundamental improvement that needs more emphasis, i.e. what the authors call the "iterative refinement", partially inherent to the SIRT method, which is in literature only achieved once for every (sub)tomogram, for different reasons: in 10 this is a design choice, in 12 and 13 this is inherent to the reconstruction technique (respectively FBP and SART, where in the latter a single iteration covers the whole subtomogram dataset).

Answer: *As proposed by reviewer #2, we have attempted to more explicitly highlight the novelty and importance of our iterative (bootstrapping) method. The following changes were added to our manuscript:*

This work presents an approach, where the DVF evolution is iteratively estimated along with mutual displacements of the measured projections and the sample reconstruction itself. In other words, the sample reconstruction is updated in each iteration given the information about projection displacement and the DVF estimation from the previous iteration. This bootstrapping iterative approach enables convergence to a consistent solution satisfying all measured projections [...]

Although the processes shown in these three references are definitely not less complex (I would even say the process shown in 12 is more complex than the RIC shown in this paper), it is possible that due to this less frequent update these methods are more prone to sudden events.

Answer: *Burning of wooden matches is undoubtedly a highly complex deformation process. However, due to the very high acquisition speed of up to 7 full tomograms per second, presented in Ruhland et al (Ref 12), the relative changes between the subsequent tomograms are of limited complexity and amplitude, as authors of Ruhland et al show in Fig 4d,e. On the other hand, we have demonstrated that our method allows accounting for a significant deformation during a single tomogram.*

As mentioned, I believe this is very nice work and it is a very good paper that deserves publication in Nature Communications, albeit with implementation of some more detailed discussion on the literature (notable 10,12, 13)

Answer: *We thank to reviewer #2 for expressing his support for our manuscript and hope that with the clarifications above and additions to our manuscript we have addressed sufficiently all of their concerns.*

Reviewer #3 (Remarks to the Author):

The authors have answered all questions convincingly. I particularly appreciate that they will disclose the code.

I am still sceptical about deducing information from a tomographic reconstruction where severe changes occur due to radiation damage, but this does not stand against using this method for more relevant problems with intrinsic dynamics. I recommend publication as is.

Answer: *We thank to reviewer #3 for their positive assessment of our work. We agree that severe radiation damage can irreversibly destroy sample small features. We have gone to great lengths to clarify that our method in this case is limited to the radiation induced change (RIC) regime, in which we observe overall deformation of the sample. The preservation of small features in our experimental result is proven by the repeatability of small features shown by the Fourier shell correlation curve. We have adequately pointed out this limitation*

in the introduction of our manuscript:

“Severe radiation damage can ultimately destroy the imaged features but already a radiation dose significantly below the maximum tolerable dose leads to deterioration of the reconstruction quality due mainly to deformation of the sample. In order to distinguish these two cases, we will be using the term radiation induced change (RIC) for the latter case. RIC in a 3D structure can be approximated to first order as a nonrigid deformation process that does not depend on the time of the scan but rather on the total deposited X-ray dose. “

REVIEWERS' COMMENTS:

Reviewer #2 (Remarks to the Author):

The manuscript can be published as is. I want to thank the authors for their adaptations and congratulate them with this nice work.